# Efficient *L*_p_ Distance Computation Using Function-Hiding Inner Product Encryption for Privacy-Preserving Anomaly Detection

**DOI:** 10.3390/s23084169

**Published:** 2023-04-21

**Authors:** Dong-Hyeon Ryu, Seong-Yun Jeon, Junho Hong, Mun-Kyu Lee

**Affiliations:** 1Department of Computer Science and Engineering, Pohang University of Science and Technology, Pohang 37673, Republic of Korea; ydh9516@gmail.com; 2Department of Computer Engineering, Inha University, Incheon 22212, Republic of Korea; roland.korea@gmail.com; 3Department of Electrical and Computer Engineering, University of Michigan-Dearborn, Dearborn, MI 48128, USA; jhwr@umich.edu

**Keywords:** functional encryption, anomaly detection, mean *p*-powered error

## Abstract

In Internet of Things (IoT) systems in which a large number of IoT devices are connected to each other and to third-party servers, it is crucial to verify whether each device operates appropriately. Although anomaly detection can help with this verification, individual devices cannot afford this process because of resource constraints. Therefore, it is reasonable to outsource anomaly detection to servers; however, sharing device state information with outside servers may raise privacy concerns. In this paper, we propose a method to compute the $L_{p}$ distance privately for even p>2 using inner product functional encryption and we use this method to compute an advanced metric, namely *p*-powered error, for anomaly detection in a privacy-preserving manner. We demonstrate implementations on both a desktop computer and Raspberry Pi device to confirm the feasibility of our method. The experimental results demonstrate that the proposed method is sufficiently efficient for use in real-world IoT devices. Finally, we suggest two possible applications of the proposed computation method for $L_{p}$ distance for privacy-preserving anomaly detection, namely smart building management and remote device diagnosis.

## 1. Introduction

Anomaly detection is defined as the identification of abnormal data, events, or phenomena that deviate significantly from common observations and cannot be described using any well-known notion of normal behavior [1]. Since its first appearance as an intrusion detection system in 1987 [2], anomaly detection has evolved in several fields, including statistics, medicine, manufacturing processes, machine learning, cyber security, and computer vision.

In particular, in Internet of Things (IoT) ecosystems, where a large number of IoT devices are connected and communicate with each other and third-party servers, it is crucial to verify whether each device operates appropriately [3]. IoT devices may malfunction because of initial faults during the manufacturing process, software bugs, mechanical failures of parts during operation, extreme changes in the surrounding environment, and malicious attacks by attackers. In real-world scenarios, an IoT ecosystem comprises of servers and several IoT devices, and it is reasonable to assume that servers are responsible for detecting such anomalous states in IoT devices because most IoT devices do not have sufficient computational resources to detect failures and recover from abnormal states. In this setup, all the normal-state information of each device, such as power consumption, network connections, and temperature, should be maintained in safe storage, and servers may periodically check whether the current state of each device is significantly different from the stored normal state. *Mean absolute error* and *mean squared error* are commonly considered to evaluate the difference (and similarity) between two state vectors. However, in 2018, Shalyga et al. [4] showed that the mean *p*-powered error is more effective in predicting short-term anomalies when *p* is greater than two on the SWaT dataset [5]. Here, the mean *p*-powered error is defined as $\frac{1}{n}\sum_{i=1}^{n}{\left|x_{i}-y_{i}\right|}^{p}$, where $x=(x_{1},\ldots ,x_{n})$ and $y=(x_{1},\ldots ,x_{n})$ are the normal state vector and the current state vector, respectively, with dimension *n*. According to the experiments in [4], the best choice for *p* is p=6.

Meanwhile, privacy concerns regarding IoT devices have arisen over the past decades as IoT ecosystems rapidly expand to industries, cities, houses, and personal wearable devices [6,7,8]. Sensor data are highly detailed, precise, and personal. Furthermore, the combination of data from multiple sensors may allow attackers to infer additional information more accurately, which is not possible with a single sensor [9]. In 2017, Nesa et al. showed that the occupancy of a room can be determined by fusing data, such as humidity, temperature, illumination level, and CO_2_ [10] with very high accuracy (up to 99.09%). This prediction becomes more precise as the number of fusion parameters increases. Therefore, privacy protection must be considered in IoT ecosystems. A recent controversy at Carnegie Mellon University over super-sensing IoT devices called Mites demonstrated potentially serious privacy concerns with IoT-based smart building management systems [11].

To protect the privacy of IoT devices, all the information collected from IoT devices must be encrypted before being transmitted to the anomaly detection server. In addition, it is desirable that an anomaly detection process should be conducted on the encrypted data instead of the original data. Furthermore, the encryption operations must be sufficiently lightweight to be run on IoT devices.

To satisfy the aforementioned requirements, we consider functional encryption (FE); we provide a formal definition of FE in the following section. FE is a special type of encryption algorithm that satisfies the following properties: let *m* be a plaintext and E(*m*) be a ciphertext of *m*. Given E(*m*), an individual with additional information, that is, a secret key bound to a function *f*, can only obtain f(m), whereas they cannot learn the original plaintext *m*. Inner product encryption (IPE) is a special form of FE whose function is the inner product, and both the ciphertext and secret key are associated with vectors. Given a secret key for the coefficient vector y and ciphertext for the plaintext vector x, we can obtain only the inner product values 〈x,y〉 through decryption but no further information about x. We say that there is a function-hiding property when information about y is protected, as well as that about x. In 2018, Kim et al. [12] proposed a practical function-hiding IPE (FHIPE) scheme. In [12] and several related works, such as [13], FHIPE was used to compute vector-related metrics, such as $L_{1}$ distance and $L_{2}$ distance, in a privacy-preserving manner. These metrics have been used for biometric authentication, nearest neighbor (NN) searches on encrypted data, and secure linear regression. However, the general $L_{p}$ distance (p>2), which is necessary for evaluating the mean *p*-powered error for advanced anomaly detection [4], has not been considered in previous studies.

In this study, we generalize the $L_{p}$ distance over the FHIPE scheme as a new distance metric for privacy-preserving anomaly detection, where p>2 denotes an even number. To be precise, our problem statement is as follows: Given a ciphertext E(x) for a vector $x=(x_{1},\ldots ,x_{n})$ and a secret key bound to a vector $y=(x_{1},\ldots ,x_{n})$, compute the $L_{p}$ distance, $\sum_{i=1}^{n}{\left|x_{i}-y_{i}\right|}^{p}$ for even *p*, using the FHIPE scheme (the mean *p*-powered error computation needs an additional operation, division by *n*. For simplicity, we omit this throughout the paper, as its computation is trivial.). Our experimental results indicate that the computation of this new metric in the ciphertext domain is efficient and applicable to highly resource-constrained IoT devices. We also suggest two applications using our method for privacy-preserving anomaly detection, i.e., smart building management and remote device diagnosis.

### 1.1. Related Works

Several applications, including biometric authentication, location-based services, data mining, and anomaly detection, encode data into vectors and then measure the distance or similarity between the vectors [13,14,15,16,17,18,19,20,21,22,23,24,25,26,27,28,29]. The vectors used in these applications may contain sensitive information. Therefore, extensive research has been conducted to measure the distance between two vectors while maintaining data privacy. Wang et al. [17] and Im, Jeon, and Lee [16] have proposed methods that encrypt a vector representing a facial image using homomorphic encryption (HE) and calculated the Euclidean distance between the encrypted vectors without decrypting them. Kang et al. [30] have shown that HE can be used to outsource a service that remotely detects whether a component of an industrial machine has diverted from its normal operating condition, without disclosing the private information of the data owner. Several previous studies used FE instead of HE. Kim et al. [12] proposed a biometric authentication method based on Hamming distance and nearest neighbor search based on $L_{2}$ distance using FHIPE. Zhou and Ren [14] proposed a biometric authentication method, called Passbio, that supports both Hamming distance and Euclidean distance. Their method does not reveal the actual distance, but only indicates whether the distance is within a predefined threshold. Kwon and Lee [31] commented on two vulnerabilities of Passbio and proposed a countermeasure against these vulnerabilities. Lee et al. [15] proposed an efficient biometric authentication method specialized for Hamming distance. Li and Jung [18] used HE and FE to encrypt geometric locations. The aforementioned applications use the Euclidean distance to evaluate the similarity between two vectors. In other applications, distance metrics, such as the Hamming distance, cosine similarity, and Mahalanobis distance, have also been used [29]. For example, the data-mining applications proposed in [20,21,22,32] calculated the cosine similarity in a privacy-preserving manner using HE, secure multiparty computation, and oblivious transfer. In studies on anomaly detection, HE and differential privacy were used to construct privacy-preserving anomaly detection systems. Alabdulatif et al. [23,24] proposed cloud-based anomaly detection systems for IoT using HE. Mehnaz and Bertino [25] also proposed a cloud-based anomaly detection system using HE. They adopted the edge computing approach in one of the stages comprising anomaly detection and used additive HE for secure aggregation of individual values from edge devices. Lyu et al. [26] proposed a privacy-preserving collaborative anomaly detection method with differential privacy. However, to the best of our knowledge, this study is the first work on general $L_{p}$ distance calculations in a privacy-preserving manner.

### 1.2. Contributions

We propose a method to compute the $L_{p}$ distance for an even number p>2 over FHIPE. We use this $L_{p}$ distance to compute the mean *p*-powered error for anomaly detection.To demonstrate the feasibility for IoT ecosystems, we implement the proposed method in C++ and conduct experiments on a prototype system composed of a desktop computer (as a server) and the Raspberry Pi (as an IoT device) system. The experimental result shows that the proposed method is sufficiently efficient to be applied to IoT ecosystems in terms of execution times and memory usage.We present two possible applications of the proposed $L_{p}$ distance computation method for privacy-preserving anomaly detection, i.e., smart building management and remote device diagnosis. Accordingly, we suggest a protocol involving multiple IoT devices and two servers.

In the remainder of this paper, Section 2 describes the preliminaries of the proposed work. Section 3 explains the proposed method on how to compute the $L_{p}$ distance over FHIPE for an even number *p*. Section 4 provides the performance analysis. Section 5 describes a general framework for privacy-preserving anomaly detection for IoT systems. Section 6 discusses the limitation of the proposed method and presents a future research direction. Conclusions are given in Section 7.

## 2. Preliminaries

### 2.1. Barreto–Naehrig Curve (BN Curve)

If computations in the extension $F_{r^{k}}$ of the prime field $F_{r}$ are feasible, an elliptical curve defined over $F_{r}$ is known as a pairing-friendly curve [33]. Specifically, the elliptical curve over $F_{r}$ should be non-supersingular and contain a subgroup whose embedding degree *k* is not considerably large. Barreto and Naehrig proposed a method for constructing pairing-friendly elliptical curves with an embedding degree k=12 [33]. This is can be expressed as: $E\left(F_{r}\right):y^{2}=x^{3}+b$ for nonzero *b*. For t≠0, they parameterized the order *q* of the elliptical curve group and the characteristic *r* as follows:(1)$$q=36t^{4}+36t^{3}+18t^{2}+6t+1,$$
(2)$$r=36t^{4}+36t^{3}+24t^{2}+6t+1.$$

### 2.2. Cryptographic Pairing

Let $G_{1}$ and $G_{2}$ be additive groups of prime order *q* and $g_{1}$ and $g_{2}$ be generators of $G_{1}$ and $G_{2}$, respectively. Subsequently, a mapping function is defined as $e:G_{1}\times G_{2}\to G_{T}$, which is used to map the elements of $G_{1}$ and $G_{2}$ onto $G_{T}$, which is a multiplicative group of prime order *q*. $0_{G_{1}}$, $0_{G_{2}}$, and $1_{G_{T}}$ denote the identity elements of $G_{1}$, $G_{2}$, and $G_{T}$, respectively. We define the map *e* as a *cryptographic pairing* when the tuple (*q*, $G_{1}$, $G_{2}$, $G_{T}$, *e*) satisfies the following properties [34]:The map *e* and group operations in $G_{1}$, $G_{2}$, and $G_{T}$ are efficiently computable.The map *e* is bilinear for all $g_{1}\in G_{1}$, $g_{2}\in G_{2}$ and *a*, *b*∈$Z_{q}$. That is,
(3)$$e\left(\left[a\right]g_{1},\left[b\right]g_{2}\right)=e{\left(g_{1},g_{2}\right)}^{ab}.$$The map *e* is nondegenerate for $g_{1}\neq 0_{G_{1}}$, $g_{2}\neq 0_{G_{2}}$. That is,
(4)$$e\left(g_{1},g_{2}\right)\neq 1_{G_{T}}.$$

Generally, cryptographic pairings that satisfy these three properties consist of group operations on elliptical curves and finite fields. Specifically, $G_{1}$ and $G_{2}$ are asymmetric, that is, $G_{1}\neq G_{2}$, and they are elliptical curve subgroups of prime order *q*. $G_{T}$ is a subgroup of the finite field $F_{r^{k}}$ for the embedding degree *k*, and its order is *q*. In this study, we use a cryptographic pairing on the BN curve, and a bilinear environment is defined as a tuple $\left(q,G_{1},G_{2},G_{T},g_{1},g_{2},e\right)$ [34]. For notational convenience, we also define a pairing-product operation. Given $P=\left(P_{1},P_{2},\ldots ,P_{n}\right)\in G_{1}^{n}$ and $Q=\left(Q_{1},Q_{2},\ldots ,Q_{n}\right)\in G_{2}^{n}$, the product of the pairings for the two vectors P and Q is defined as follows:(5)$$e_{prod}\left(P,Q\right)=\Pi_{i=1}^{n}e\left(P_{i},Q_{i}\right).$$

### 2.3. FHIPE

In this study, we used the FHIPE scheme $\prod_{IPE}$ proposed by Kim et al. [12]. $\prod_{IPE}$ is defined as (IPE.Setup,IPE.KeyGen,IPE.Encrypt,IPE.Decrypt). The dimensions *n* of vectors x and y are given as parameters, and *S* is a subset of $Z_{q}$, which specifies the possible range of the inner product.



$IPE.Setup(1^{\lambda },S)$

1.Select a bilinear environment $\left(q,G_{1},G_{2},G_{T},g_{1},g_{2},e\right)$ according to the security parameter λ.2.Choose a matrix $B\leftarrow {\mathrm{GL}}_{n}\left(Z_{q}\right)$, where ${\mathrm{GL}}_{n}\left(Z_{q}\right)$ refers to a group of n×n square matrices whose elements belong to the finite field $Z_{q}$ and an inverse matrix exists.3.Compute $B^{★}\leftarrow \det\left(B\right)\cdot {\left(B^{-1}\right)}^{⊤}$.4.Output the public parameter $pp=(q,G_{1},G_{2},G_{T},e,S)$ and master secret key $msk=(pp,g_{1},g_{2},B,B^{★})$.

IPE.KeyGen(msk,y)

1.Choose a uniformly random element $\alpha \overset{R}{\leftarrow }Z_{q}$.2.Using the master key msk and vector $y\in Z_{q}^{n}$, output the secret key $sk=\left(K_{1},K_{2}\right)=\left(\left[\alpha \cdot \det\left(B\right)\right]g_{1},\left[\alpha \cdot y\cdot B\right]g_{1}\right)$ such that $K_{2}\in G_{1}^{n}$.

IPE.Encrypt(msk,x)

2.Choose a uniformly random element $\beta \overset{R}{\leftarrow }Z_{q}$.2.Using the master key msk and vector $x\in Z_{q}^{n}$, output the ciphertext $ct=\left(C_{1},C_{2}\right)=\left(\left[\beta \right]g_{2},\left[\beta \cdot x\cdot B^{★}\right]g_{2}\right)$ such that $C_{2}\in G_{2}^{n}$.

IPE.Decrypt(pp,sk,ct)

1.Using the public parameter pp, secret $sk=(K_{1},K_{2})$, and ciphertext $ct=(C_{1},C_{2})$, compute $D_{1}=e\left(K_{1},C_{1}\right)$ and $D_{2}=e_{prod}\left(K_{2},C_{2}\right)$.2.Find a solution z∈S for $D_{1}^{z}=D_{2}$. If *z* exists, it is the inner product of x and y, denoted as z=〈x,y〉. Output *z* if it exists; otherwise, output ⊥, which indicates that no solution exists.

### 2.4. $L_{1}$ and $L_{2}$ Distances over FHIPE

Kim et al. [12] proposed an encoding method based on the FHIPE scheme to compute the $L_{1}$ distance (Hamming distance) between two binary vectors. Given two binary vectors $x,y\in {\left\{0,1\right\}}^{n}$, vectors x and y are encoded as $x^{\prime }$ and $y^{\prime }$$\in {\left\{-1,1\right\}}^{n}$ by changing the zeroes in x and y to −1. Subsequently, the Hamming distance between x and y is $d\left(x,y\right)=\left(n-\langle x^{\prime },y^{\prime }\rangle \right)/2$.

Jeon and Lee [13] used the square of the $L_{2}$ distance (squared Euclidean distance) as a metric of similarity in a face authentication system. They designed three operations, namely, EncodeX, EncodeY, and Euclid, using the encoding method proposed by Kim et al. [12]:EncodeX(msk,x)1.Construct an (n+2)-dimensional vector $x^{\prime }={(∥x∥}^{2},-2x_{1},-2x_{2},\cdots ,-2x_{n},1)$ from $x=(x_{1},\ldots ,x_{n})$.2.Output $ct=IPE.Encrypt(msk,x^{\prime })$.EncodeY(msk,y)1.Construct an (n+2)-dimensional vector $y^{\prime }=(1,y_{1},y_{2},\cdots ,y_{n}{,∥y∥}^{2})$ from $y=(y_{1},\ldots ,y_{n})$.2.Output $sk=IPE.KeyGen(msk,y^{\prime })$.Euclid(pp,sk,ct)1.Calculate z=IPE.Decrypt(pp,sk,ct).2.Output *z*. *z* satisfies $z=\langle x^{\prime },y^{\prime }\rangle ={(∥x∥}^{2}-2x_{1}y_{1}-2x_{2}y_{2}-\cdots -2x_{n}y_{n}+{∥y∥}^{2}{)=(∥x∥}^{2}-2\langle x,y\rangle +{∥y∥}^{2}{)=∥x-y∥}^{2}.$

## 3. Proposed Method

We present a method for computing the $L_{p}$ distance over FHIPE for an even number *p*. Technically, we compute the *p*-powered $L_{p}$ distance. However, computing the $L_{p}$ distance as the *p*-th root from the *p*-powered $L_{p}$ distance is trivial.

Given two vectors $x\in Z_{q}^{n}$ and $y\in Z_{q}^{n}$, the *p*-powered $L_{p}$ distance can be calculated as follows:(6)$$L_{p}^{p}\left(x,y\right)={∥x-y∥}_{p}^{p}=\sum_{i=1}^{n}{\left|x_{i}-y_{i}\right|}^{p}=\sum_{i=1}^{n}{\left(x_{i}-y_{i}\right)}^{p},$$
where the last equality comes from the fact that *p* is even. For the two *n*-dimensional vectors u and v, the ⊙ operator denotes *element-wise* multiplication, i.e., $u⊙v=(u_{1}v_{1},\ldots ,u_{n}v_{n})$, and $v^{\left(t\right)}$ denotes the *t*-th *Hadamard power* of v for any t≥0, i.e., $v^{\left(0\right)}=\left(1,\ldots ,1\right)$ and $v^{\left(t\right)}=v^{\left(t-1\right)}⊙v$ for t≥1 [35]. Note that $L_{p}^{p}\left(x,y\right)$ in (Equation 6) is expressed as a sum of *p*-th powers of binomial $\left(x_{i}-y_{i}\right)$. Therefore, we can apply the binomial theorem, xi−yip=∑j=0ppjxij(−yi)p−j=∑j=0ppj(−1)p−jxijyip−j, to each term ${\left(x_{i}-y_{i}\right)}^{p}$. Consequently, Lpp(x,y)=∑i=1n∑j=0ppj(−1)p−jxijyip−j, and we obtain:(7)Lpp(x,y)=∑i=0ppi(−1)p−i〈x(i),y(p−i)〉.

We encode x and y to $x^{\prime }\in Z_{q}^{(p-1)n+2}$ and $y^{\prime }\in Z_{q}^{(p-1)n+2}$ over a ring $Z_{q}$ as follows:(8)$$x^{\prime }=(\alpha_{0}{∥x∥}_{p}^{p},\alpha_{1}x^{\left(p-1\right)},\alpha_{2}x^{\left(p-2\right)},\cdots ,\alpha_{p-1}x^{\left(1\right)},\alpha_{p}),$$
(9)$$y^{\prime }=(1,y^{\left(1\right)},y^{\left(2\right)},\cdots ,y^{\left(p-2\right)},y^{\left(p-1\right)}{,∥y∥}_{p}^{p}),$$
where αi=pi(−1)i for i=0,…,p. Thus, we observe that:(10)$$L_{p}^{p}\left(x,y\right)=\langle x^{\prime },y^{\prime }\rangle .$$

Accordingly, a generalized form of the *p*-powered $L_{p}$ distance computation protocol over FHIPE is defined as $\prod_{PDIST}$=(Setup, EncodeX, EncodeY, GetDistance), which is obtained from using $\prod_{IPE}$ for a security parameter λ∈N, degree p∈2N, the dimension n∈N of the input vectors, and the range *S* of the inner product. These four operations are defined as follows:$Setup(1^{\lambda },S)$1.Select a bilinear environment $\left(q,G_{1},G_{2},G_{T},g_{1},g_{2},e\right)$ according to the security parameter λ.2.Choose a matrix $B^{\prime }\leftarrow {\mathrm{GL}}_{l}\left(Z_{q}\right)$ for l=(p−1)n+2, where ${\mathrm{GL}}_{l}\left(Z_{q}\right)$ is a group of l×l square matrices whose elements belong to the finite field $Z_{q}$ and an inverse matrix exists.3.Compute ${B^{\prime }}^{★}\leftarrow \det\left(B^{\prime }\right)\cdot {\left(B^{\prime }-1\right)}^{⊤}$.4.Output the public parameter $pp^{\prime }=\left(G_{1},G_{2},G_{T},q,e,S,p\right)$ and master secret key $msk^{\prime }=\left(pp^{\prime },g_{1},g_{2},B^{\prime },{B^{\prime }}^{★}\right)$.$EncodeX(msk^{\prime },x)$1.Construct a vector $x^{\prime }\in Z_{q}^{l}$ as described previously (Equation 8).2.Output $ct^{\prime }=IPE.Encrypt\left(msk^{\prime },x^{\prime }\right)$.$EncodeY(msk^{\prime },y)$1.Construct a vector $y^{\prime }\in Z_{q}^{l}$ as described previously (Equation 9).2.Output $sk^{\prime }=IPE.KeyGen\left(msk^{\prime },y^{\prime }\right)$.$GetDistance(pp^{\prime },sk^{\prime },ct^{\prime })$1.Calculate $z=IPE.Decrypt(pp,sk^{\prime },ct^{\prime })\in S$.2.Output *z*. *z* satisfies $z=\langle x^{\prime },y^{\prime }\rangle =L_{p}^{p}\left(x,y\right)$.

## 4. Performance Analysis

To verify the feasibility of the proposed method, we implemented $\prod_{PDIST}$ in C++ using the FHIPE library developed in a previous study [13] (our algorithm and code are available at https://github.com/inhaislab/FHIPE-Lp-Distance (accessed on 19 April 2023)). As shown in Table 1, we also used NTL 11.3.2 [36] for integer, vector, and matrix arithmetic, MCL 1.51 [37] for pairing-based cryptography, and GMP 6.1.2 [38] for big number arithmetic. We applied $\prod_{PDIST}$ to our testing programs on desktop PC and Raspberry Pi. The programs were run on a desktop computer with 16 GB main memory, an AMD Ryzen5 3600 6-Core processor @ 3.60 GHz, and Ubuntu 20.04 (using Windows 11 Pro WSL2), and on a Raspberry Pi 2 model B with 1 GB RAM and a quad-core ARM Cortex-A7 processor @ 900 MHz. MCL 1.51 [37] provides various parameter sets for bilinear environments. Among these, we selected *SNARK1*, which is one of the most frequently used parameter sets for Ethereum.

We measured the execution times for the four operations of $\prod_{PDIST}$ with n={8,16,32,64,128} and p={2,4,6,8,10}. For each combination, 100 measurements were averaged. Figure 1 shows the experimental results on the desktop PC. The exact numerical data are given in the Appendix A (See Table A1, Table A2, Table A3 and Table A4.) Based on the results, it can be deduced that Setup and GetDistance are the dominant operations. In particular, the execution time for GetDistance increases exponentially as *p* increases. For example, it took 0.01 s, 0.07 s, 0.56 s, 6.07 s, 57.3 s for p=2,4,6,8,10, respectively, when *n* is 64. Notably, the main operation in GetDistance is the discrete logarithm (DL) computation. The complexity of solving a DL problem is $O(\sqrt{S})$ when a well-known general method is used, that is, the baby-step giant-step method. It must be noted that *S*, which is the range of the solution of the DL problem, satisfies $S=n\cdot m^{p}$, where *m* is the maximum difference between the two vector elements $x_{i}$ and $y_{i}$. This explains the exponential growth in execution time with an increase in *p*. We also observe that the time increases slowly as *n* increases because the execution time is proportional to $\sqrt{n}$. Setup also consumes a significant amount of time, for example, 6.65 s on average when *n* = 64 and *p* = 10, because it involves complex matrix operations, such as inverse computation. However, EncodeX and EncodeY consume negligible time compared with Setup and GetDistance. For example, they only took 24.77 ms and 21.80 ms, respectively, when *n* = 64. This is because they involve relatively lower cost operations, such as matrix-vector multiplication and elliptical-curve point multiplication.

We also measured the execution time on the Raspberry Pi. Setup and GetDistance were not considered in this experiment because these operations are expected to run not on IoT devices but on high-end servers. Thus, the implementation on an IoT device, that is, Raspberry Pi, requires only EncodeX and EncodeY. Figure 2 shows the experimental results for these operations. The exact numerical data are provided in the Appendix A. (See Table A5 and Table A6.) The overall tendencies of EncodeX and EncodeY on the Raspberry Pi were similar to those of the previous experiment on the desktop PC. Between the two operations, EncodeY required slightly shorter execution times than EncodeX on both the desktop and Raspberry Pi, up to approximately 33% on the desktop and approximately 50% on the Raspberry Pi. Notably, even a combination of the largest parameters, that is, (n,p)=(128,10), requires only a few seconds for execution. To be precise, the execution times were 3.82 s and 2.58 s for EncodeX and EncodeY, respectively. Under typical settings with p=6 [4], the time required was less than a second.

Finally, we compared the proposed method with previous privacy-preserving anomaly detection methods. In particular, we focused on the distance metric to identify anomaly. Table 2 demonstrates that all the previous privacy-preserving anomaly detection methods were based on either Euclidean distance ($L_{2}$ distance) or absolute error ($L_{1}$ distance). Even though a customized nonlinear metric based on Gaussian distribution was used in [25], it was computed on plaintext. Therefore, it cannot be directly compared with the proposed method. By examining the distance metric as a quantitative property, we can see that the proposed method is the first anomaly detection method that performs distance computation over ciphertext using the $L_{p}$ distance metric for p>2.

## 5. Applications

In this section, we propose a general framework for privacy-preserving anomaly detection in IoT systems. We present two examples of this framework: smart building management and remote device diagnosis. Our basic assumption is that each IoT device has several normal states defined by its specifications. Each of these states can be represented as a latent vector, that is, a feature vector, by applying a neural network model as a feature extractor. The normal-state vectors are stored, and anomaly detection is performed by comparing the current state with the normal states. An anomaly is defined as an event in which the current feature vectors are not sufficiently close to any of the stored normal feature vectors in terms of the $L_{p}$ distance metric. We used $\prod_{PDIST}$ as a building block for privacy-preserving anomaly detection.

As shown in Figure 3,the framework is composed of two servers and several IoT devices. $Server_{setup}$ denotes the set up server. We assume that the normal-state vectors $x_{1},\ldots ,x_{M}$ are provided to $Server_{setup}$ as a specification of each target device. In the device registration stage, $Server_{setup}$ performs Setup for that device, encrypts $x_{1},\ldots ,x_{M}$ into $ct_{1}^{\prime },\ldots ,ct_{M}^{\prime }$ using EncodeX with the corresponding $msk^{\prime }$ of the device, and transfers the results to an anomaly detection server $Server_{detect}$ for registration. $Server_{setup}$ stores $msk^{\prime }$ for each device, and $Server_{detect}$ stores $ct_{1}^{\prime },\ldots ,ct_{M}^{\prime }$ for each device. When a device must be analyzed for an anomaly, it obtains temporal access to $msk^{\prime }$ in $Server_{setup}$ after it passes proper authentication. It encrypts the state vector y into $sk^{\prime }$ using EncodeY and sends the results to $Server_{detect}$ with a request for an anomaly detection service. Upon receiving this request, $Server_{detect}$ determines whether the device is currently in a normal state based on an NN search using the $L_{p}$ distance. Thus, $Server_{detect}$ computes $GetDistance(pp^{\prime },sk^{\prime },ct_{i}^{\prime })$ for i=1 to *M* for this device and checks whether at least one of these distances is smaller than a predefined threshold. Otherwise, an anomaly is asserted.

To verify the feasibility of the above-mentioned application framework, we conducted additional experiments on the Secure Water Treatment (SWaT) dataset [5]. SWaT is a time series anomaly dataset provided by iTrust of Singapore University of Technology and Design (SUTD). The data in SWaT were collected using 24 sensors and 27 actuators on the water treatment testbed over 11 days through six steps: water storing, pre-treatment and chemical dosing, fine filtration, dechlorination, removing inorganic impurities, and water storing for distribution. Therefore, the SWaT dataset can be interpreted as a 51-dimensional time series dataset. Each time series is composed of numerical sensor and actuator values measured every second. We can model the water treatment application considered in the SWaT dataset using the components in Figure 3 as follows: the $Server_{setup}$ and $Server_{detect}$ are the third party participants that are in charge of detecting anomalies in the entire water treatment system, and each IoT device in Figure 3 represents a water treatment plant. In each individual water treatment plant, a feature vector is created through a neural network-based feature extractor. Feature extraction is essentially a dimensionality reduction process. For this purpose, we used an encoder with three hidden fully connected layers of dimensions 4n, 2n, and *n*, respectively. This encoder can be viewed as the encoding layers of an autoencoder. Using an autoencoder is a well-established approach in the literature in time series anomaly detection [39]. We considered a hyperparameter *n* to represent the size of the final latent vectors, i.e., feature vectors. We tested n=8,16, and 32. This *n* corresponds to the vector size, *n*, in the experiment in Section 4. Among the 51 sensors and actuators, we removed six unstable data columns as in [39]. To represent the short-term temporal behavior of each sensor and actuator, we considered 60 consecutive measurements, i.e., 60 s for each sensor or actuator. Consequently, an input for the feature extractor consists of 45 time series with each series containing 60 measurements. In other words, a single input is a 60 × 45 matrix and the output is an *n*-dimensional feature vector. The feature vector is encrypted through the EncodeX and EncodeY operations for registration and anomaly detection, respectively.

For our experiment, we used the same experimental setup as that in Section 4 (see Table 1). The feature extractor was implemented in C++ to extract feature vectors. Then, each IoT device conducted feature extraction followed by EncodeX (or EncodeY). The experimental results measured at the anomaly detection stage, i.e., the time for feature extraction plus EncodeY is presented in Figure 4. According to the results, the time required for feature extraction was roughly on the same order as that for EncodeY. Note that the total computation time on Raspberry Pi was only 0.6396 s even for the combination of largest hyperparameters, that is, p=10 and n=32.

### 5.1. Smart Building Management

The first application scenario involves smart building management. As already discussed in Section 1, IoT-based smart building management systems can cause serious privacy issues if the collected data are not properly protected [11]. We assume that $Server_{setup}$ belongs to the facility management department in a campus comprising several buildings, and $Server_{detect}$ belongs to another department. Managers should be responsible for managing the IoT devices of their department, which is accomplished by observing the measured status.

An IoT device must be registered in advance using the following procedure. The device manager requests that the facility management department register the IoT device. The department then issues a unique device id $device_{i}$ and conducts Setup for the device to create its $msk_{i}^{\prime }$. Referring to the device specifications, $Server_{setup}$ obtains the normal-state vectors $x_{1},\;\ldots ,\begin{array}{c} x_{M} \end{array}$ and encrypts them into $ct_{i1}^{\prime },\;\ldots ,ct_{iM}^{\prime }$ using EncodeX with $msk_{i}^{\prime }$. The manager then stores $ct_{i1}^{\prime },\ldots ,ct_{iM}^{\prime }$ in the department database for $Server_{detect}$.

Subsequently, anomaly detection is performed as follows: a manager who wants to investigate a suspicious $device_{i}$ in their department requests temporal access from $msk_{i}$ to $Server_{setup}$. After authenticating this request, $Server_{setup}$ sends ($msk_{i}^{\prime }$, sessionId) to $device_{i}$ and ($device_{i}$, sessionId) to $Server_{detect}$. Device $device_{i}$ executes EncodeY using $msk_{i}^{\prime }$ with the current feature vector $y_{i}$ to output $sk_{i}^{\prime }$. Subsequently, $device_{i}$ sends ($device_{i}$, sessionId, $sk_{i}^{\prime }$) to $Server_{detect}$ in the department. $Server_{detect}$ validates the session and solves an encrypted NN search problem to obtain the minimum mean *p*-powered error by calculating $GetDistance(pp^{\prime },sk_{i}^{\prime },ct_{ij}^{\prime })$ for j=1, …,M. If the error is below a certain threshold, the device is considered healthy.

### 5.2. Remote Device Diagnosis

The second application scenario is remote device diagnosis. We assume that an IoT appliance manufacturing company provides self-diagnosis services to its clients and that the company knows the normal states of its devices before release. The company configures $Server_{setup}$ and $Server_{detect}$ such that all $msk^{\prime }$s for the devices are stored in $Server_{setup}$ and not shared with $Server_{detect}$ and ensures that all the encrypted normal-state vectors of each device are stored in $Server_{detect}$.

When a user of $device_{i}$ wants to know whether their IoT appliance is working well, the user allows the device to initiate an anomaly detection protocol in a manner similar to that described in Section 5.1. The device pulls $msk_{i}^{\prime }$ from $Server_{setup}$ and uses $msk_{i}^{\prime }$ to encrypt the current state in $ct_{i}^{\prime }$ and sends it to $Server_{detect}$. Finally, the device displays the result received from $Server_{detect}$ to the user.

### 5.3. Possible Attacks and Mitigation on Our Systems

In this subsection, we address the possible attacks against each component of our system.

Attack to devices: an attacker may attempt to extract either the state vector or $msk^{\prime }$ by observing the memory of a device while encryption is performed with $msk^{\prime }$. However, this type of physical threat can be mitigated using a a trusted execution environment.Network attack: attackers may attempt to sniff the communication between a device and servers or steal the normal-state vectors stored in the database. As the vectors are encrypted by EncodeX and EncodeY, attackers learn no information about the vectors, even when they obtain the encrypted vectors. Attackers may attempt to perform replay attacks and man-in-the-middle attacks on the communication between a device and servers to manipulate data. These attacks can be prevented by marking with a timestamp and authenticating with a message authentication code or digital signature of the server on every request and response.Attack to servers: if $Server_{setup}$ is compromised, $msk^{\prime }$s for all devices may be leaked. Therefore, we assume that $Server_{setup}$ is protected with a proper mechanism. We also assume that $Server_{setup}$ is trustworthy. In other words, it does not attempt to recover the device information with $msk^{\prime }$. Under this assumption, the only concern involves $Server_{detect}$, which may want to recover any useful information from its database of encrypted feature vectors. (This also includes the case where $Server_{detect}$ is compromised from outside attackers.) However, this is prevented by the security property of FHIPE. For this, however, $Server_{setup}$ and $Server_{detect}$ must be strictly separated to ensure that they do not share $msk^{\prime }s$ with each other.

## 6. Limitation

The proposed method works only when the degree *p* is even. In (Equation 6), the $L_{p}$ distance is composed of terms that are *p*-powers of $|x_{i}-y_{i}|$. Since it is not possible to compute an absolute value using FHIPE, we replaced $|x_{i}-y_{i}{|}^{p}$ with ${\left(x_{i}-y_{i}\right)}^{p}$. This is only possible when *p* is even. To be precise, $|x_{i}-y_{i}{|}^{p}=\left(\right|x_{i}-y_{i}{{|}^{2})}^{p/2}={\left({\left(x_{i}-y_{i}\right)}^{2}\right)}^{p/2}={\left(x_{i}-y_{i}\right)}^{p}$. However when *p* is odd, this equality does not hold. To resolve this issue, we may consider a naive method for odd *p* as follows: given a maximum difference *m* between two vector elements $x_{i}\in Z_{q}$ and $y_{i}\in Z_{q}$, we can compute $x_{i}-y_{i}$ by computing the inner product of $\left(x_{i},-1\right)$ and $\left(1,y_{i}\right)$ using FHIPE. If $x_{i}-y_{i}\leq m$, then $|x_{i}-y_{i}|=x_{i}-y_{i}$. Otherwise, if $x_{i}-y_{i}>m$, this implies that $y_{i}$ is actually greater than $x_{i}$ and $x_{i}-y_{i}=q-m^{\prime }$ for $y_{i}-x_{i}=m^{\prime }\leq m$. Therefore, the actual absolute value can be computed as $q-\left(x_{i}-y_{i}\right)=m^{\prime }$. However, the above approach requires FHIPE Setup, KeyGen, Encrypt, and Decrypt for each absolute value term $|x_{i}-y_{i}|$ separately. This may raise both performance and security issues. Therefore, a more advanced solution is required for odd *p*, which will be a future research direction.

## 7. Conclusions

In this study, we developed a method to compute the $L_{p}$ distance privately using FHIPE for even p>2. The proposed method is the first anomaly detection method that performs distance computation over ciphertext using the $L_{p}$ distance metric for p>2. Our experimental results demonstrate that the proposed method is sufficiently efficient for use in real-world IoT devices. For example, EncodeX and EncodeY require only 787 ms and 430 ms, respectively, for a typical setting with p=6 and n=64 on a Raspberry Pi IoT device according to the experimental results in Section 4. Based on additional experiments using the SWaT dataset in Section 5, the time required to extract a 32-dimensional feature vector from a 60 × 45 input matrix using a three-layer fully connected neural network is 0.2604 s. If a typical setting is considered with p=6 [4], the time for EncodeY on the Raspberry Pi side is 0.1944 s and the time for GetDistance on the server side is 0.4222 s. Therefore, the total computation time for anomaly detection is (0.2604+0.1944+0.4222M) s when there are *M* possibilities for normal states. Assuming that *M* is not very large, e.g., M<10, the overall anomaly detection operation can be completed in a few seconds in a typical setting with p=6. Therefore, the $L_{p}$ distance can be used to compute the mean *p*-powered error for anomaly detection without revealing the actual state of IoT devices. Even though the anomaly detection server computes the distance between encoded state vectors of an IoT device, it never sees the actual values of these vectors. Thus, the proposed method is expected to be useful for outsourcing anomaly detection in a privacy-preserving manner. We suggest two practical scenarios for privacy-preserving anomaly detection: a smart building management system and remote device diagnosis.

## Figures and Tables

**Figure 1 sensors-23-04169-f001:**
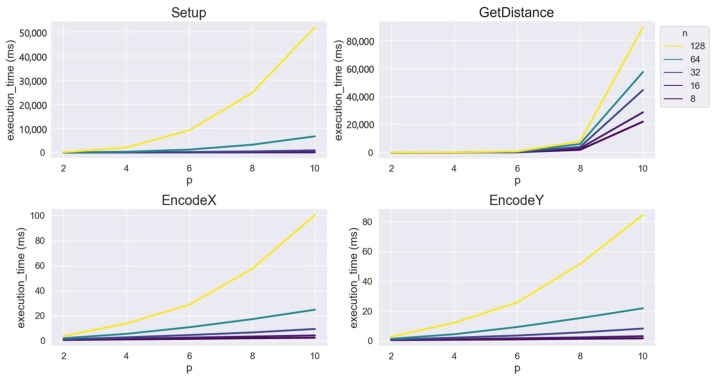
Execution time (ms) of the operations of $\prod_{PDIST}$ on a desktop PC.

**Figure 2 sensors-23-04169-f002:**
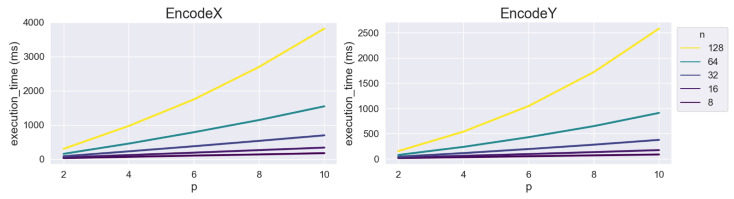
Execution time (ms) of EncodeX and EncodeY on Raspberry Pi.

**Figure 3 sensors-23-04169-f003:**
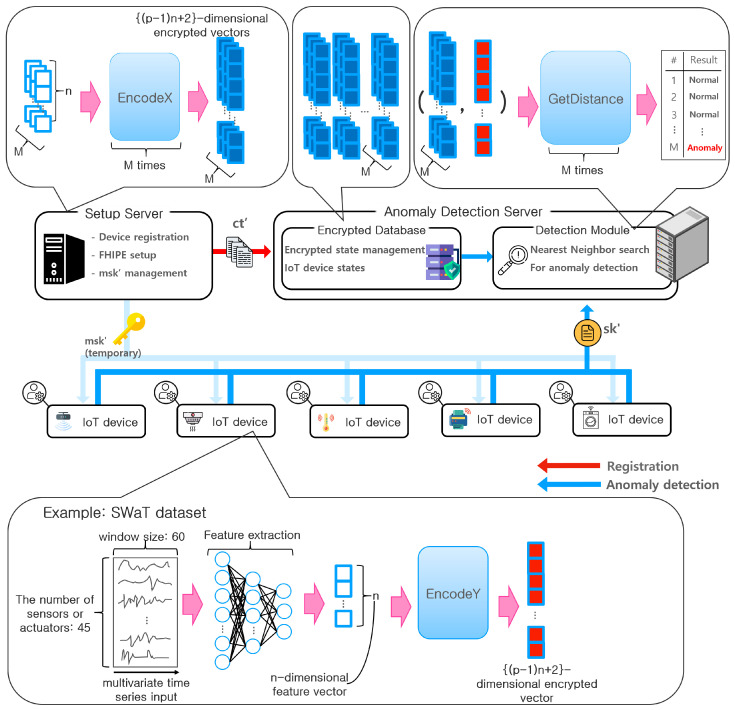
Example framework for privacy-preserving anomaly detection.

**Figure 4 sensors-23-04169-f004:**
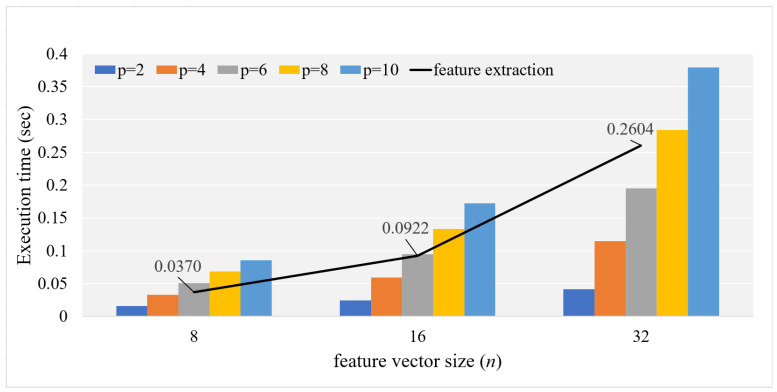
Total execution time for feature extraction and EncodeY in Raspberry Pi. The line and bars represent the execution times of feature extraction and EncodeY operations, respectively.

**Table 1 sensors-23-04169-t001:** Experimental setup for performance analysis.

	Desktop Computer	Raspberry Pi
CPU	AMD Ryzen5 3600 6-Core @ 3.60 GHz	ARM Cortex-A7 4-Core @ 900 MHz
Memory	16 GB	1 GB
OS	Ubuntu 20.04 (using Windows 11 Pro WSL2)	Raspbian
language	C++11	
SW library	FHIPE library [13], NTL 11.3.2 [36], MCL 1.51 [37], GMP 6.1.2 [38]	

**Table 2 sensors-23-04169-t002:** Comparison of privacy-preserving anomaly detection methods (* computed in plaintext).

Method	Cryptographic Primitive	Protection	Metric	*p*
[23]	HE	Client data	Euclidean distance	2
[24]	HE	Client data	Euclidean distance	2
[25]	HE (Additive only)	Client data	Q-function * (customized metric)	–
[26]	Differential Privacy	Train data	Mean absolute error *	1
Proposed	FE	Client data	$L_{p}$ distance	any even *p*

## Data Availability

The data presented in this study are openly available at https://github.com/inhaislab/FHIPE-Lp-Distance (accessed on 19 April 2023).

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
