# Peer review of "Efficient Lp Distance Computation Using Function-Hiding Inner Product Encryption for Privacy-Preserving Anomaly Detection"

_sensors, 2023, doi:10.3390/s23084169_

Round 1
Reviewer 1 Report
The results obtained in the paper are publishable, subject to some necessary changes. The techniques used to solve the problem are standard with some novelty, and the results obtained are correct. However, there are some points need to be further clarified before its final acceptance for publication: 1. The motivation on the study should be further emphasized, particularly; the main advantages of the results in the paper comparing with others should be clearly demonstrated. 2. English should be further improved, since the paper has some spelling and grammar errors. Also the paper has some editing problems. 3. The example section needs to be further expanded and including some remarks to show the effectiveness and efficiency of the proposed method, compared with others. 4. Some remarks on Theorem 1 would be necessary and helpful. 5. The authors should consider update references, and the following recently published papers on this general topic could be helpful authors to make some comments and comparisons
Reviewer 2 Report
The paper proposes a method to compute the Lp distance privately for even p>2 using inner product functional encryption and use this method to compute an advanced metric, that is, p-powered error, for anomaly detection in a privacy-preserving manner.
The paper is well written and can be accepted.
The authors can consider the following suggestion.
1) Specify what is S on page 4, line 127.
2) The authors have consider the value of p to be even. Kindly, mention what would happen when p is odd.
3) It would have been easier to see proof of the concept, if the algorithm and codes are made public.
Reviewer 3 Report
The paper is very interesting. It has good content and it is well written. However, there are some minor issues that need to be addressed before submission:
Here are a few minor suggestions for the article "Efficient Lp Distance Computation using Function-Hiding Inner Product Encryption for Privacy-Preserving Anomaly Detection":
1. Clarify the problem statement: In the introduction section, it would be helpful to clarify the specific problem you are addressing with this research. This will give readers a better understanding of the context and motivation behind your work.
2. Provide more details about the data: It would be useful to provide more information about the type of data you are working with (e.g., numerical data, text data, image data, etc.), as this will help readers understand how your approach can be applied in different contexts.
3. Clarify the use case: In the section where you describe the use case for your approach, it would be helpful to provide more details about the specific scenario in which this approach would be useful. This will help readers understand the practical implications of your work.
4. Add more discussion about related work: While you do mention related work in the introduction, it would be helpful to include a more detailed discussion of related work in the related work section. This will help readers understand how your approach compares to existing approaches and what the unique contributions of your work are.
5. Add more information about the experimental setup: In the experimental section, it would be useful to provide more information about the experimental setup (e.g., the specific datasets used, the hardware and software used, etc.). This will help readers understand the limitations of your approach and the extent to which your results can be generalized to other scenarios.
6. Cite the following recent publication: CorrAUC: a Malicious Bot-IoT Traffic Detection Method in IoT Network Using Machine Learning Techniques.
Reviewer 4 Report
In this paper, the authors proposed anomaly detection for the privacy of IoT devices to operate effectively. They indicated that experimental results are promising and satisfy the requirements for the systems. Also, the authors suggested applications used in the smart building for privacy-preserving anomaly detection. Using p >2 calculations, test results show that the computation of this new metric in the ciphertext domain satisfies the requirement. These can be applied to IoT devices that consume high resources.
The article’s topic is interesting and novel. But supporting experiments are not enough to verify the research.
There are some concerns and criticisms that should be addressed:
1) The application section is not clear. The authors mentioned about the neural network model as a feature extractor in this part. However,
the explanation of the model is not enough, and Figure 1 and Figure 2 should be explained more clearly by comparing the execution times mentioned in the figure.
2) 5.1 Experiments set in the Smart Building Management part are not defined. Figure 3 is a general picture and does not reflect the work done in this research.
3) Experiments setup and results are not enough to support the proposed methodology. There should be more clear ways to explain the experiments and results to verify the model.
4) In conclusion, the authors stated that the proposed method is expected to be useful for outsourcing anomaly detection in a privacy-preserving manner. How is it useful?
Compare the findings with other results by using quantitate numbers.
5) Why is the proposed model sufficiently efficient for real-world IoT devices?
6) The limitations of the work need to explain in more detail.
7) The conclusion should be improved more and give some comparison results including the numbers.
